# Clinical Applications of Poly-Methyl-Methacrylate in Neurosurgery: The In Vivo Cranial Bone Reconstruction

**DOI:** 10.3390/jfb13030156

**Published:** 2022-09-19

**Authors:** Tomaz Velnar, Roman Bosnjak, Lidija Gradisnik

**Affiliations:** 1Department of Neurosurgery, University Medical Centre Ljubljana, 1000 Ljubljana, Slovenia; 2AMEU-ECM Maribor, 2000 Maribor, Slovenia; 3Laboratory for Cell Cultures, Medical Faculty Maribor, 2000 Maribor, Slovenia

**Keywords:** biopolymers, in vivo reconstruction, cranial bone, poly-methyl-methacrylate, bone flap

## Abstract

Background: Biomaterials and biotechnology are becoming increasingly important fields in modern medicine. For cranial bone defects of various aetiologies, artificial materials, such as poly-methyl-methacrylate, are often used. We report our clinical experience with poly-methyl-methacrylate for a novel in vivo bone defect closure and artificial bone flap development in various neurosurgical operations. Methods: The experimental study included 12 patients at a single centre in 2018. They presented with cranial bone defects after various neurosurgical procedures, including tumour, traumatic brain injury and vascular pathologies. The patients underwent an in vivo bone reconstruction from poly-methyl-methacrylate, which was performed immediately after the tumour removal in the tumour group, whereas the trauma and vascular patients required a second surgery for cranial bone reconstruction due to the bone decompression. The artificial bone flap was modelled in vivo just before the skin closure. Clinical and surgical data were reviewed. Results: All patients had significant bony destruction or unusable bone flap. The tumour group included five patients with meningiomas destruction and the trauma group comprised four patients, all with severe traumatic brain injury. In the vascular group, there were three patients. The average modelling time for the artificial flap modelling was approximately 10 min. The convenient location of the bone defect enabled a relatively straightforward and fast reconstruction procedure. No deformations of flaps or other complications were encountered, except in one patient, who suffered a postoperative infection. Conclusions: Poly-methyl-methacrylate can be used as a suitable material to deliver good cranioplasty cosmesis. It offers an optimal dural covering and brain protection and allows fast intraoperative reconstruction with excellent cosmetic effect during the one-stage procedure. The observations of our study support the use of poly-methyl-methacrylate for the ad hoc reconstruction of cranial bone defects.

## 1. Introduction

The employment of new technological achievements and the number of invasive techniques in medicine has been increasing in recent years, enabling us to perform more invasive and complex procedures, which is particularly important in the surgical field [1,2]. This progress, however, has not been possible without the advances and improvements in basic science and in the new technology that had paved the path. One of the main goals of modern medicine remains the renewal of lost or damaged tissues and the promotion of healing mechanisms, leading to the best possible restitution of the tissue [1]. The process of regeneration is very complex; it is not complete and may not be achievable in many instances during healing. Instead, reparation takes place, replacing the functional tissue with fibrous tissue [2]. Both reparation and regeneration may be influenced by numerous factors, interfering with the healing processes and eventually leading to a poor healing outcome [1,2].

To promote the healing mechanisms, numerous practices have been exploited in clinical medicine, including the techniques of tissue augmentation [3,4,5,6,7,8,9]. These techniques are very effective for stimulating the regeneration and augmentation of bone [10,11,12]. In a variety of neurosurgical operations, bone defects are very frequently encountered and a difficult problem to cope with, both in paediatric and in adult surgery, for example, in tumour infiltration of skull bone, cranial base, reconstruction operations, fibroplasia and trauma. In such cases, the reconstruction of the skull bone is problematic due to tissue deficit [13,14,15,16].

Many alternatives exist for skull bone reconstruction, including autografts, allografts or artificial replacement material [17,18]. Each of them has its advantages and disadvantages, as well as proper clinical indications. In large defects that impede bone healing and cause functional and aesthetic difficulties, artificial materials may be successfully used to replace the missing bone, such as metal, aliphatic polyesters, hydroxyapatite and metilmetacrilate, which is the most widely used material in cranioplasty [16,19,20].

In clinical practice, especially in extensive or repeated neurosurgical procedures, including trauma, tumours and a wide range of resections, incomplete covering due to the bone defects of various aetiologies may lead to numerous postoperative complications [21,22]. Not only is the bone defect cosmetically problematic, but it also poses a problem for the brain tissue, which remains unprotected under the skin flap. Besides aesthetic and protective problematics, the craniectomies have also been linked to functional deficits, such as postoperative meningocele and the syndrome of the trephined [23,24]. Moreover, it has been documented that the cerebrospinal fluid (CSF) leak and associated difficulties were more common after craniectomy even in fully closed dura and skin and that the late craniectomy reconstruction can be potentially difficult due to the fibrosis attaching the skin flap on the dural or brain surface [25,26,27].

Many materials may be used for bone regeneration and as artificial bone substitutes in repairing bone defects, especially the cranial bones. These materials include metal and other artificial bone graft materials, such as polymers [28,29,30,31]. When considering the artificial bone substitutes, some are osteoconductive and are thus largely applied in reconstructions of large bone defects, such as calcium phosphate-based biomaterials (including hydroxyapatite, ceramics and calcium phosphate cement). Other materials, such as recombinant human bone morphological proteins, are osteoinductive and are valuable for promoting fracture healing [29,32]. In cranial bone reconstruction practice, besides the osteointegration and the biological properties, the artificial flap needs to be unresorbable and needs to satisfy also other roles, such as brain protection and aesthetic function, thus substituting the cranial bone. For such purposes, metal (titanium, tantalum, titanium and their alloys, magnesium and its alloys), ceramic, poly-methyl-methacrylate, poly-L-lactic acid (PLA), polylactide (PLLA) and polyetheretherketone (PEEK) are most often used [29,32,33,34]. As state-of-the-art, 3D-printed bioactive glass is gaining importance in clinical research [33,35]. All of them have their advantages and disadvantages in clinical practice. Metallic materials and suitable for repairing large bone defects in load-bearing areas; they are mechanically strong and biologically inert. Ceramic exhibits high compressive strength and abrasion resistance, and allows tissue ingrowth, therefore securing a permanent connection between the implant and tissue. It can be manufactured as a bioceramic-based scaffold with integrative antibacterial and osteogenic functions, providing significant benefits over biological counterparts [36,37,38]. PEEK is polymeric material with high structural strength and stable physicochemical properties. Its advantages include wear-, fatigue-, corrosion resistance and biologically safety [28,33,39].

No uniform strategies have been recommended so far as to the techniques of closing the bone defect in neurosurgical practice. The choice is left to the surgical team regarding the technique and the materials of choice [21,22]. These bone defects may be reconstructed using a range of artificial or natural materials, which are either custom-made (industrially 3D-produced implants) or manufactured during the operation directly or in vivo. For this purpose, poly-methyl-methacrylate is often used. This material has many advantages making it very useful not only for neurosurgery but for surgical practice in general [11,21,40].

In the article, we describe the poly-methyl-methacrylate properties and share our clinical experience with poly-methyl-methacrylate for a novel in vivo bone defect closure in various neurosurgical operations.

### The Properties of Poly-Methyl-Methacrylate

Poly-methyl-methacrylate was first introduced into the industry in the 1930s for manufacturing aeroplane windows and canopies [40,41]. Since then, its composition has been modified according to medical applications [42,43]. For cranioplasty, poly-methyl-methacrylate is one of the most widely used materials. Its medical use started for repairing cranial bone defects, firsts experimentally in monkeys and then in 1941 in humans [44]. From then on, poly-methyl-methacrylate is a standard material for such purposes. Besides neurosurgery, it is also vastly used in dentistry, maxillofacial surgery, traumatology and orthopaedics.

The poly-methyl-methacrylate is a dough-like material, which is formed by mixing two components, consisting of polymer powder and a liquid monomer [45]. The first one comprises a combination of pre-polymerized poly-methyl-methacrylate beads, a radiopaque element (BaSO_4_ or ZrO_2_ particles) and an initiator (benzoyl peroxide) for the polymerization reaction of the free radicals. The powder also contains an inorganic radio-pacifying agent, usually barium sulphate (10–15 wt %), which is used for better visualisation during clinical use, and an antibiotic [34]. The liquid component is composed of methyl-methacrylate monomer, an activator (N, dimethyl-para-toluidine), acting as an activator of the formation of radicals, and a small quantity of hydroquinone. The latter is used as an inhibitor of the reaction or a stabilizer, preventing premature polymerization during storage [45,46,47,48]. A colouring agent such as chlorophyll may be added as an option [48]. In cement products, the amounts of their constituents may vary and this influences the cement properties. Moreover, antibiotics and sometimes vitamin E, chitosan and silver nanoparticles can be added into the mixtures due to their antibacterial activity, thus reducing the postoperative infection rate [48,49,50]. Before the implantation, both components are mixed. Mixing under a vacuum may be performed in some instances to reduce the material porosity and enhance its mechanical properties [47].

During the implant preparation in the in vivo setting, both components are mixed at room temperature. For clinical use, there are four different stages during poly-methyl-methacrylate cement preparation. According to the cement viscosity: (I) mixing phase, (II) waiting phase, (III) working or application phase and (IV) hardening or adjusting phase. The mixing phase is the time for perfect homogenization of the liquid monomer and the polymer powder phases. It may take up to one minute. The homogeneity of the mixing poly-methyl-methacrylate cement is affected by using many parameters, such as the number of strokes, the design of the mixing of the spatula and vessel and the mixing speed or revolutions per minute. More forceful and prolonged mixing of the cement makes it more porous. The waiting phase is the time for attaining a non-sticky cement, which can be prepared for the application. It may take up to some minutes and depends on the cement type and the handling temperature. During the working phase, the cement is utilised in contact with live tissue (bone) before the prosthesis implantation. This phase takes two to four minutes and may vary according to the temperature of handling and the type of cement [46,51,52]. The hardening or adjusting phase is the final period of the polymerisation process, which takes one to three minutes. The reaction is exothermic, heating the surrounding tissue to 70 °C. Care must be therefore taken when manufacturing and placing a poly-methyl-methacrylate bone flap to the cranial defect in order not to damage the underlying brain tissue. The temperature generated by the polymerization drops back to the ambient temperature at the end of the hardening phase. As the cement is prepared with a certain mixing ratio in weight percentage, the first step is to find a suitable mixing ratio of the solid and liquid components. The increase in powder quantity results in difficulty in mixing the components. Further, the setting and curing time differs with various component ratios [46]. During polymerization, it is possible to form the amorphous mass according to the desired shape to fit the bone defect. The material hardens and cools in several minutes and can then be safely positioned in place. When the polymerization is completed, poly-methyl-methacrylate becomes very hard. Although drilling and cutting are possible, this is best performed during the polymerization phase when the material is still soft [52,53,54].

## 2. Materials and Methods

The main purpose was to provide an alternative and novel technique for in vivo cranial bone reconstruction with poly-methyl-methacrylate during neurosurgical operations. The prospective experimental study was started in the neurosurgery centre in Ljubljana in 2018. Altogether, twelve patients were included in the experimental study. There were seven men and five women. The mean age of the patients was 48 years, range 29 to 75 years. The study involved patients, who presented with cranial bone defects after neurosurgical procedures, including (I) tumour, (II) traumatic brain injury and (III) vascular pathologies.

The tumour group included meningioma patients where the tumour invaded and destructed the skull bone and the reuse of the bone flap was therefore impossible. Here, the cranial reconstruction was performed immediately after tumour resection. The tumour group encompassed five patients, two men and three women.

The trauma patients underwent craniotomy due to cerebral oedema or large subdural hematoma, needing decompression with craniectomy. This group of patients was operated on due to the undelaying pathology and the reconstruction was performed three to six months after the initial insult. The trauma group included four patients, three men and one woman.

The vascular pathology group comprised the patients after ruptured aneurysm or arteriovenous malformation surgery, where the repositioning of the bone flap was impossible due to cerebral oedema. This group involved three patients, one man and two women, in whom the cranial reconstruction was performed after the brain oedema receded.

The neurosurgical part of the operation in all patients included a standard procedure, depending on the cause of surgery. The missing bone was reconstructed from poly-methyl-methacrylate immediately after the tumour removal in the tumour group (Figure 1). The groups of trauma and vascular patients required a second surgery for cranial bone reconstruction when cerebral oedema subsided. Patients were followed up through regular outpatient department visits. The first outpatient visit was performed in the trauma group after three to six months and in the vascular group after three months.

Other inclusion criteria encompassed the following: adult patient population (older than 18 years), no secondary brain tumour (the absence of malignant tumours), obligatory destruction and invasion of the cranial bone by tumour, destruction of the cranial bone because of trauma (bone fragmentation beyond reconstruction), the position of the bone defect on the convexity of the cranial vault, positive prevention swab of the bone flap (at our centre, the swab is always taken when removing the bone flap during the operation), no previous oncological treatment (especially irradiation) and a good anticipated treatment outcome for all three groups of patients. The patients with malignant brain tumours and meningiomas involving the cranial base were excluded, as well as those with the anticipated bone defect extending on the cranial base and intraorbital area, with the bifrontal craniectomy, and heavily neurologically deteriorating patients (traumatised patients with both unresponsive pupils for more than four hours, Hunt Hess 5 patients in the vascular group and patients older than 80 years) [55,56].

### The Technique for Cranial Reconstruction

The artificial bone flap was modelled in vivo from two-component poly-methyl-methacrylate material. We have utilised Palacos R + G, a high viscosity, radiopaque bone cement with gentamicin (Heraeus Medical, Wahrheim, Germany). The artificial flap was moulded and modelled according to the shape of the removed bone just before the closure of the skin. In patients with convexity meningiomas invading the cranial bone, which was therefore not useful for coverage, the original cranial bone flap was used as a template. The bone was placed on sterile paper and the paper template was formed. The sterile paper was used to wrap the bone flap before the application of the poly-methyl-methacrylate to prevent the sticking of the material to the bone flap surface. The thickness of the original bone flap was assessed before the modelling with the calliper. The two components of the poly-methyl-methacrylate were mixed and left to dry for a few minutes (Figure 2). Then, the mixture gained a correct consistency, similar to the modelling clay and did not stick to the surgical gloves. The poly-methyl-methacrylate flap was modelled according to the paper template and its thickness was adjusted to the original bone flap (Figure 3). Since the poly-methyl-methacrylate is soft during the polymerisation phase, it can be modelled as wished, thinned, bent and shaped according to the desired form. The excess material was cut off with scissors, as the poly-methyl-methacrylate was still soft. The thickness of the artificial flap was compared both to the original bone flap, which was removed, and to the healthy cranial bone. In addition, the curvature of the artificial flap was adjusted according to the original one to fit the bone defect optimally. During the hardening, the temperature of the poly-methyl-methacrylate flap rose (it reaches 90 °C) and it was left for a few minutes to cool. The drilling for the dural suspension was performed with a bone drill and when ready, the poly-methyl-methacrylate flap was fixed to the bone defect with craniofixes or with self-drilling screws and titanium plates (Figure 4).

In the trauma and the vascular group, the cranial bone flap was removed during surgery and stored for later use. Our practice is that the bone is always deeply frozen and stored in the −80 °C freezer. Such bone flaps are normally implanted some months later, when the brain oedema subsides completely and when the patients start to recover. However, some shrinkage of the bone flap may occur during storage and in some cases, the control swabs taken during the craniectomy operation may be positive. In these instances, the in vivo modelling of the artificial flap is necessary. In our patients from the trauma and the vascular group, who for the above reasons could not receive their original bone flap and required such intervention, a poly-methyl-methacrylate flap was employed. It was modelled during the reconstructive phase of the operation. The surgical procedure was the same as outlined above, using a template of sterile paper, which was shaped according to the bone defect.

The patients were treated in the neurosurgical ward according to the standard clinical practice. The next day after the reconstructive operation, a control computer tomography (CT) scan was performed on all patients as a control.

## 3. Results

In all twelve patents included in the experimental study, cranial bone reconstruction was needed due to the bone defect, which was irreplaceable by the original cranial bone. The reasons for cranial bone removal were traumatic brain injury, tumour infiltration of the bone and vascular pathology with the underlying brain oedema. In all patients, the poly-methyl-methacrylate flap was modelled in the operation theatre, either in the course of the initial operation (i.e., in the tumour group) or during the second operation, which was performed three to six months after the initial insult.

The tumour group included five patients with meningiomas destroying the bone of the cranial vault. In three patients, the brain was oedematous because of the tumour growth and brain invasion, which did not affect the course of bone reconstruction and further recovery. Antioedematous therapy with dexamethasone for a few days successfully helped in the oedema subsidence, not affecting the neurological status of the patients. As the tumour was growing through the dura mater into the cranial bone, the dural excision was also needed and replaced by an artificial substitute (dural patch). In one patient, only a partial dural removal was possible, which did not affect the postoperative treatment course and potential tumour relapse. No cerebrospinal fluid (CSF) leak was noted. Where the meningioma was extending through the cranial bone, invading the subcutaneous tissue of the galea, the tumour was divided from the macroscopically healthy subcutaneous tissue and excised completely. All bone flaps were cut in such a way, that a safety margin was encircling the tumour completely. The poly-methyl-methacrylate flap was manufactured during surgery, substituting the infiltrated bone completely, meaning that the patients needed no further reconstructive surgery. The hospital stay ranged from five to seven days and no neurological sequelae were noted after the tumour operation and during the discharge.

One patient reported to the outpatient clinic four months after the cranial reconstruction. Signs of hypertrophic granulations were observed in the wound scar, together with a small amount of purulent discharge. Here, a late postoperative infection was recorded. The CT scan confirmed epidural collection and the removal of the artificial flap with wound sterilisation was needed. This artificial flap was re-sterilized and then used again. After the course of antibiotics, the wound healed completely and a later, reconstructive operation was planned and performed two months after the infection had subsided. This patient needed two postoperative surgeries, one for the removal of the infected artificial flap and another for its relocation after the regression of the infection. Detailed information for tumour patients is presented in Table 1.

The trauma group included four patients, all with severe traumatic brain injury. Because of fulminant brain oedema, decompressive craniectomy was needed, together with other invasive and intensive measures for elevated intracranial pressure management. In two patients, the cranial bone was fragmented due to high-energy trauma and was beyond reconstruction, and the dural tear was present. Here, the dural reconstruction was needed with a suture and resorbable material. After the decompressive craniectomy, the brain surface was covered with the lyophilised bovine pericardium, to prevent the adhesions between the barn and subcutaneous tissue, which may hamper the bone reconstruction later on. In this group of patients, the bone defects were large, since the decompressive craniectomy needs to be as large as possible to decompress the brain tissue sufficiently. No postoperative complications and postoperative CSF leaks were noted. The patients from the trauma group needed two operations, the first was performed at the time of admission after the brain injury and the second one was a reconstructive surgery, following after three to six months. The recovery time was long in this group due to a severe brain injury these patens sustained and was followed by long-term rehabilitation. The information for the trauma group is presented in Table 2.

The vascular group comprised three patients. One sustained a rupture of arteriovenous malformation with intracerebral bleeding. Two patients suffered a subarachnoid haemorrhage due to an aneurysm rupture on the internal carotid artery. The first patent was operated on, the intracerebral haematoma was evacuated and the arteriovenous malformation was located and excised. Due to brain oedema, decompressive craniectomy was needed. In patients with a ruptured aneurysm, endovascular treatment excluded the aneurysm completely. However, due to subarachnoid haemorrhage and subsequent brain oedema with uncontrollable intracranial pressure, decompressive craniectomy was performed at a later stage. Bone defects were large to account for an effective decompression. No postoperative complications, such as infections or CSF leaks were recorded. In this group, two surgeries were needed; a decompressive one and then a reconstructive one, after the rain oedema receded. The treatment course was long for these patients due to the nature of the underlying pathology. The complete information for the vascular group is presented in Table 3.

The average modelling time for the poly-methyl-methacrylate flap modelling was approximately 10 min, with additional 10 min for cooling. The convenient location of the bone defect, located on the skull convexity, enabled a relatively straightforward and fast reconstruction procedure. After the flap modelling and final trimming, the intraoperative fit was very good, giving as well as an excellent cosmetic result, covering the bone defect completely. The underlying dura mater or artificial dura in case of tumour patients, protecting the brain, was not severed. The fixing of the poly-methyl-methacrylate flap to the surrounding bone was unproblematic since the artificial material was soft and placing the fixing screws or craniofixes was straightforward. No deformations of flaps were encountered during their fixation to the bone. The fit was very good, giving an excellent cosmetic result as well as brain protection.

The day after the reconstruction, a control CT scan showed a good position of the implant with no displacement, deformation or fluid collection underneath. The rest of the postoperative course was uneventful. No complications were observed and no reoperation or revision was necessary, except for one patent from the tumour group, as mentioned above. At the discharge, there were no signs of cerebrospinal fluid leakage or infection and the wounds were healing by first intention.

During the follow-up at the outpatient department, three months after the reconstruction in the tumour and the vascular group and six months after the reconstruction in the trauma group, the wound was healed completely and the aesthetic result was excellent. No deformations were noticed and the patients reported no allergic reactions (Figure 5).

## 4. Discussion

This clinical study was designed to evaluate the usefulness of poly-methyl-methacrylate for the reconstruction of cranial bone defects in vivo. It included the patients after convexity meningioma operations, traumatic brain injuries and vascular operations, where the bone was removed as a result of cerebral oedema. The research was based on the experience with poly-methyl-methacrylate practice in trauma and orthopaedic surgery, where this artificial material is used as bone cement for supplementary fixing of the prosthetic material [57,58,59]. Due to the good results obtained so far, the efficacy of poly-methyl-methacrylate was also tested in neurosurgical practice.

When considering the operation of both malignant and benign tumours invading the skull bone and cranial base, many technical difficulties may arise. Because of tissue deficiency, especially the skull bone and soft tissue, the reconstruction cannot be easily accomplished, thus presenting a challenge for a surgeon, especially in later phases of operation [60,61,62,63,64,65,66]. Years ago, many skull base tumours, principally those of the anterior or middle cranial fossa and those extending into the orbital cavity, were not excised completely. Recently, improved techniques of craniofacial surgery have been developed, allowing a wide range of reconstructions and consequently leading to more successful clinical results [67]. Benign tumours, such as meningiomas, may also destroy the nearby structures. Some meningiomas are invasive and about 5% of meningiomas are malignant, more likely causing direct invasion [60,61,62,63]. In addition to tumours, other pathologies, such as traumatic brain injury and vascular pathologies, may result in bone removal, as a result of cerebral oedema that prevents the placement of the bone flap. In all cases, bone reconstruction is needed, when the initial insult has subsided. In such extensive resections, the aesthetic reconstruction of large bone defects may pose a significant issue during the operation. The tissue in the form of autografts and allografts is one attractive option. Another one is artificial replacement material. The selection of the material and operative technique depends on surgeons’ experience and preferences in addition to the size, location, shape and depth of the bone defect [57,58,59,60,61,62,63,64,65,66,67,68,69,70,71].

In our study, we have decided on an in vivo reconstruction of the missing bone with an artificial replacement material for several reasons. The bone autografts, which are available in the form of split bone, and harvested from another part of the skull, allow coverings of the large volume of tissue defects and are relatively straightforward to use. Autografts are an ideal choice for grafting procedures, providing osteoconductive scaffolds, osteoinductive growth factors and osteogenic cells. Autografts are safe in terms of disease transmission and exhibit no immune response reactions; they show a low infection and resorption rate that leads to relatively short-term graft incorporation [67,72]. On the other hand, the use of viable autologous tissue has many limitations, especially restricted graft availability, morbidity at the donor site after the operation, higher infection rate due to a more extensive operative process, longer intraoperative time and sometimes the need for the plastic surgeon. When allografts are considered, the infection risk is higher and they possess the risk of disease transmission. Additionally, immunological reactions are possible, complicating the recovery and slowing down the healing process; therefore, allografts are not so safe and difficult to obtain. Consequently, they are not frequently used [72,73,74].

Further possibilities include cranioplasty implants made of various composite materials. These alternatives for cranial bone reconstruction include porcelain, titanium mesh, titanium plate and polyetheretherketone (PEEK). Each has its advantages and disadvantages [73,75,76,77,78]. Titanium is immunologically inert, durable and firm. No adverse effects are noted when implanted in the body and it is well tolerated by the immune system. It can be fixed to the skull by screws that are placed through pre-existing holes in the artificial flap. The disadvantages include its rigidity, which means, that once formed, it is not possible to adjust it to the bone defect. This is one of its drawbacks since it is well known that the cranial bone in the region of craniectomy may be resorbed and reformed to some extent, resulting in a suboptimal fit of the titanium flap [79]. Conversely, titanium mesh is more flexible, enabling its shaping when placed to the bone defect. Fixation to the bone is straightforward. On the other hand, the mesh is very thin and can be easily broken or deformed, needing great care during the cranial reconstruction. It is also less resilient to trauma in cases of falls and head injuries, since it is very thin, in comparison to the titanium plate. Another advantage of both titanium flap and mesh is that they can also be implanted in an inflamed area; however, this is not performed in neurosurgical practice. PEEK is a high-performance semi-crystalline engineering thermoplastic with excellent mechanical strength, outstanding resistance, very low moisture uptake and good dimensional stability. The hardness is much lower in comparison to titanium. PEEK can be drilled with a high-speed bone and therefore manufactured to shape to some extent when needed. Similar to titanium, it can be sterilized and used more than once. On the other hand, the poly-methyl-methacrylate flap cannot be used more than once. In case of infection, it must be discarded and a new bone substitute needs to be used [79,80,81,82,83,84,85].

The artificial bone flaps made of titanium, PEEK and porcelain are custom-made in advance of the implantation. Before the operation, an accurate CT scan of the head is needed to manufacture a three-dimensional model for artificial flap manufacturing [83,84,85]. The CT scan is performed with 1 mm slices and with no gantry, ensuring that the flap, once produced, is completely compatible with the bone defect in the patient’s skull (Figure 6). One disadvantage is that the artificial bone flaps lack osteoinductive or osteogenic properties. Their production may take weeks before being delivered to the clinical department and it is not very available. Moreover, these products are very expensive and their production needs to be planned well in advance, according to the shape of the bone defect [72,77,81,82,83,84,85].

It has been reported that cranioplasty with autologous bone grafts is associated with high overall complication rates in comparison to bone substitutes. Among them, PEEK and poly-methyl-methacrylate have relatively low complication rates. One of the important disadvantages is an infection, which may occur with autologous bone and bone substitutes as well. In such cases, the removal of the autologous or artificial flap is needed to heal the infection properly (Figure 7). Unusually, these infections are not life-threatening and may be quickly resolved with antibiotic therapy; however, the recovery course is prolonged, and in the later phase, another cranial reconstruction is needed [84,85,86,87].

In our cases, we have decided on artificial bone reconstruction. Since custom-made cranioplasty implants need to be manufactured in advance according to the bone defect, the fit may not always be appropriate, increasing the possibility that the implant may shift. Furthermore, a second operation for the implantation is needed. Our goal was to complete the tumour resection and the reconstruction in one leg, posing fewer operation-associated risks and enabling faster recovery. Poly-methyl-methacrylate is efficient and offers a relatively straightforward procedure. Besides being performed immediately after the tumour resection, it is fast and yields excellent cosmetic results. The material is soft and it can be modelled in the shape of the removed bone, exactly filling the bone defect. It can be thinned as necessary and the thickness of the emerging artificial flap is measured with a scale during the phases of its manufacturing. During the hardening phase, it allows subtle additional adjustments in shape; thus, the curvature of the implant may easily be adjusted according to the curvature of the skull. The original cranial bone (the template) is very welcome since it provides information about the thickness and curvature of the artificial flap [88,89,90]. For that reason, we have used the original bone as a template and it was therefore not difficult to produce the correct shape of the artificial flap. The neuronavigation is of great benefit during the operation and is used for planning the resection and the safe margin location [91,92,93,94,95,96,97]. The implant may be straightforwardly fixed with titanium plates and screws, titanium clamps or absorbable clamps, providing good stability. The implant, when in place and air-dried, yields a solid construction that perfectly matches the patient’s natural head shape and has good strength in both compression and tension. Moreover, poly-methyl-methacrylate can be instantly available and relatively inexpensive. Our operative procedure with bone modelling did not pose any particular technical problems and was time-saving with a good result of the reconstruction.

One limitation with the in vivo bone reconstruction with poly-methyl-methacrylate is the shape of the reconstructed bone [88,89,93]. The skull convexities are the most suitable locations for such reconstructions. On the other hand, especially problematic are the locations at the curvatures on the cranial base, the orbit, the bottom of the temporal fossa, the front orbital plane and particularly the facial bones [88,94,98]. These very complex structures cannot be modelled freehand and therefore, other reconstructive techniques must be used. In particular, 3D-printing and computer-assisted manufacturing represent considerable benefits for the reconstruction of such complex defects and are indeed gaining importance in such cases, allowing the developing of very complex substitutes that closely fit the bone defects, making the reconstruction perfect [96,97,98,99,100,101,102]. The 3D manufacturing approach can be used conveniently for convexity areas; however, these new techniques also exhibit some drawbacks, making the in vivo bone flap modelling a suitable alternative in comparison to more complicated techniques. Besides price, 3D manufacturing requires time, first completing accurate imaging (usually a CT scan for a 3D reconstruction). Then, the manufacturing process itself is quite long. In some clinical settings and situations, it is not possible to wait for a manufactured 3D-bone flap and thus, in vivo modelling is preferable [95,99]. Not all centres have access and the possibility to use such products. Additionally, the vital cranial bone near the bone defect (at the site of craniotomy) may change with time and since 3D manufacturing may be time-consuming, the artificial flap may not fit the bone defect optimally. During the in vivo modelling, on the other hand, the poly-methyl-methacrylate flap can be adjusted and forms exactly according to the cranial bone defect [92,93,94,95].

Poly-methyl-methacrylate as a non-degradable biomaterial, is suitable due to its structural and biomechanical properties, as well as antibacterial influence. This is the reason that it can also be used for implanting in inflamed areas (as during spinal and cranial procedures involving infections and oral surgery) [88,89,90,97]. Poly-methyl-methacrylate, however, promotes cell ingrowth and is, therefore, useful as a substitution for native bone. The cells will attach to this material and will grow in its pores, resulting in a good integration with the biological tissues [91,92,93]. In our case, biodegradability is an unwanted complication, since an implanted artificial bone flap is required to remain unchanged in situ. Its function is to protect the brain tissue and to exert an acceptable cosmetic effect, thus repairing the defect of the native cranial bone after trauma, tumour or infection [101,102]. These requirements have also been met in our study, confirming the poly-methyl-methacrylate as a superior material for cranial bone substitution [94,95,96,98,99,100,101].

According to our experience, this technique is useful in appropriate conditions, as it is time-saving, straightforward, cost-effective and enables a good ad hoc bone reconstruction. It offers brain protection with a good cosmetic result after the extensive cranial vault defect at the time of tumour resection.

## 5. Conclusions

Poly-methyl-methacrylate is a suitable material, allowing fast intraoperative reconstruction with excellent brain protection and cosmetic effect during the one-stage procedure. These products are readily obtainable. Their use is time-saving, straightforward and offers good dural covering and brain protection, resulting in an optimal cosmetic effect and healing outcome. As with all artificial materials, the possibilities of infections and host versus graft immune reactions are always present, as it is a higher price and the availability of some of these products. The observations of our study support the use of poly-methyl-methacrylate for the reconstruction of cranial bone defects.

## Figures and Tables

**Figure 1 jfb-13-00156-f001:**
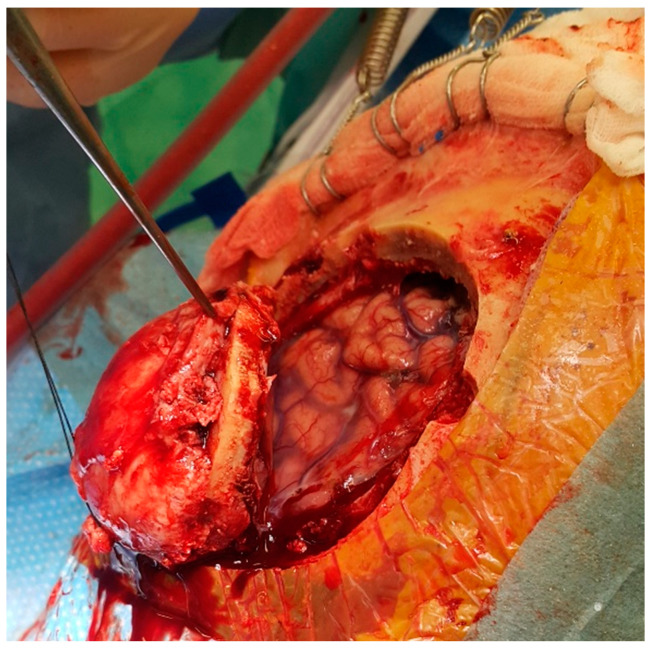
The meningioma is growing through the cranial bone, destructing it. The bone flap infiltrated with the tumour was removed. The brain underneath is compressed but not infiltrated.

**Figure 2 jfb-13-00156-f002:**
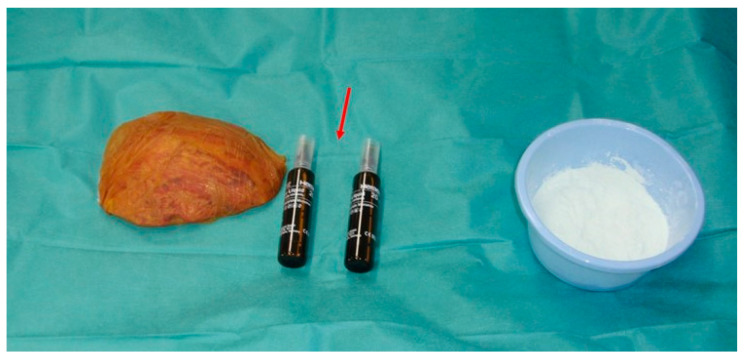
The poly-methyl-methacrylate components are sterile and individually packed. The liquid component is provided in a glass vial (arrow) and the dry component (powder) is stored in separate packaging. Both components are mixed to obtain a suitable consistency. On the left, the original bone flap is visible, which will be disposed of due to tumour invasion. A new, artificial bone flap will be formed from the poly-methyl-methacrylate.

**Figure 3 jfb-13-00156-f003:**
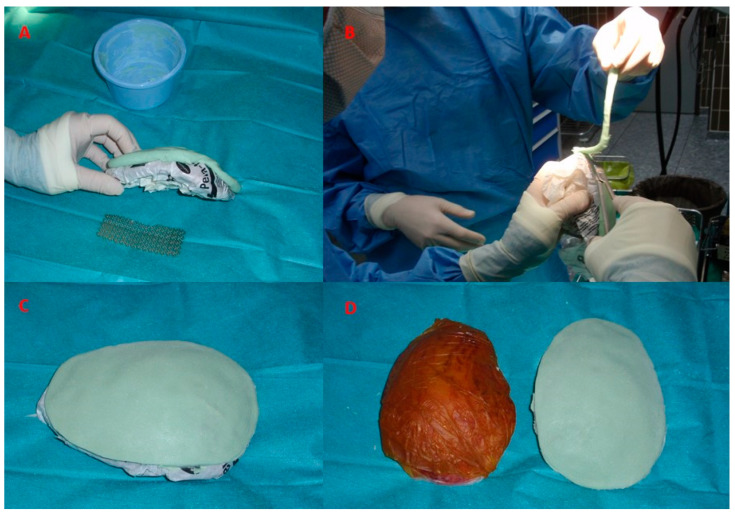
The template is designed according to the excised bone flap and the artificial substitute made of poly-methyl-methacrylate will be sculpted according to this template (**A**). While still soft, the excess material is easily removed with scissors (**B**). During the hardening phase, the curvature of the artificial flap is adjusted according to the curvature of the skull (**C**). The template and the completed poly-methyl-methacrylate flap, ready for implantation (**D**).

**Figure 4 jfb-13-00156-f004:**
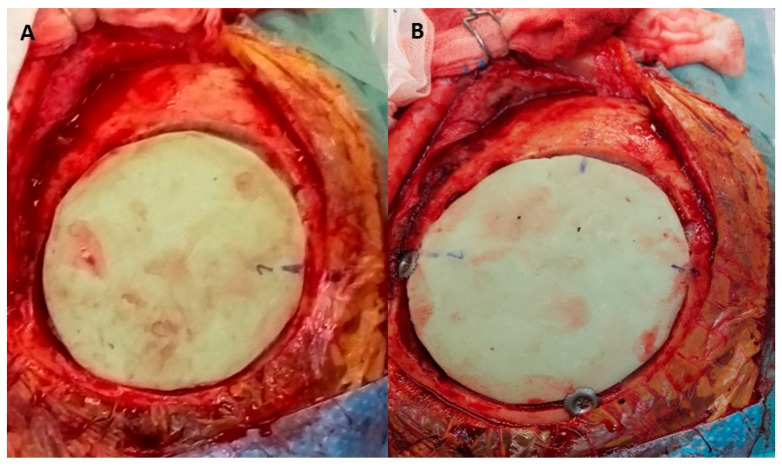
The artificial flap is placed in the bone defect and will be fixed to the native bone with craniofixes (**A**). The implantation of the artificial flap is completed. The bone defect is well covered and the fibrin glue has been applied to seal the tiny gaps between the bone and the artificial flap. It will be covered by a skin flap, which has been lifted in the early phase of the operation (**B**).

**Figure 5 jfb-13-00156-f005:**
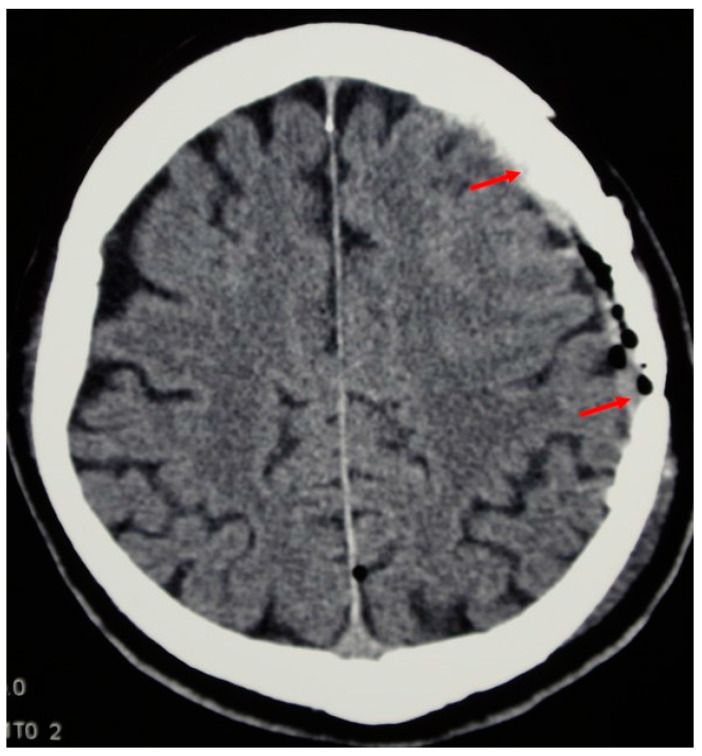
The follow-up CT scan shows the implant integration with the cranial bone (arrows denote both ends of the artificial flap).

**Figure 6 jfb-13-00156-f006:**
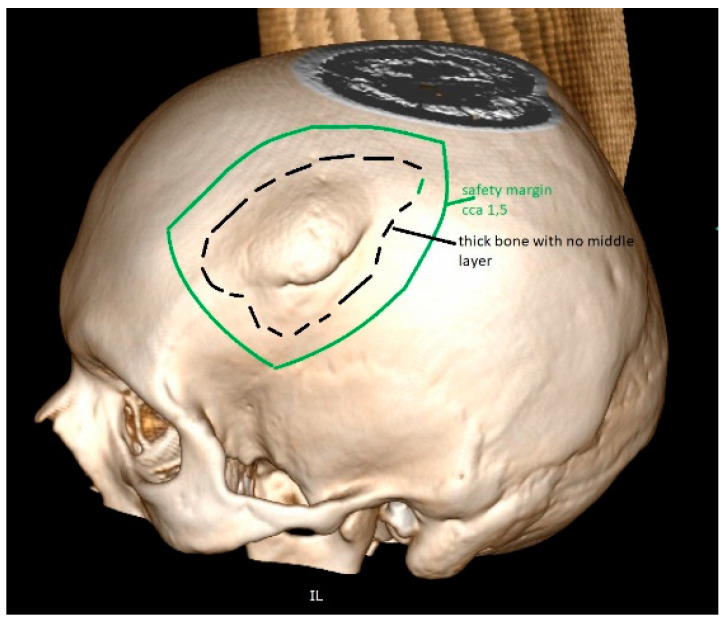
The tumour invades the cranial bone. This part will be removed during surgery. For the reconstruction, one alternative also includes PEEK, which is very popular for artificial flap manufacturing. The 3D reconstruction needs to be performed before the manufacturing of such PEEK implant. The safety margin of resection is included in the planning, providing enough space for diseased bone removal and proving a good fit.

**Figure 7 jfb-13-00156-f007:**
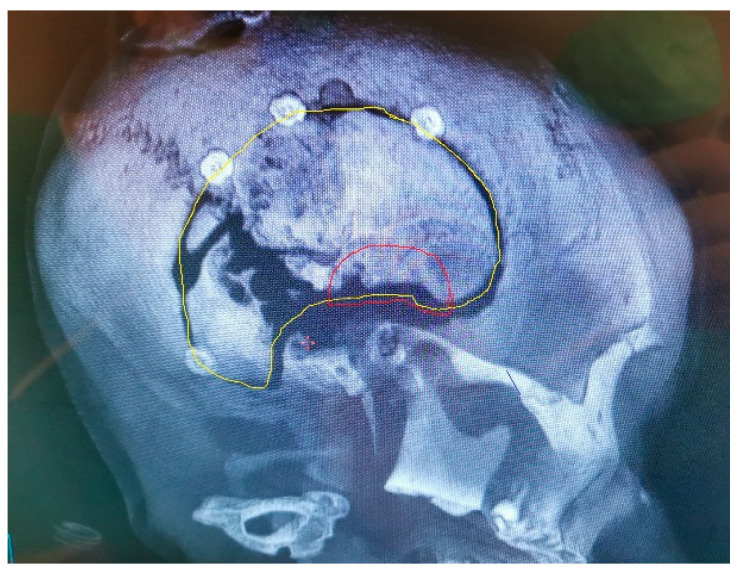
The original bone flap has been infected and partially resorbed, as documented on CT with 3D reconstruction. The craniofixes can be seen, fixing the bone flap to the skull. This infected bone flap will be surgically removed and replaced by a new one made of poly-methyl-methacrylate, that was manufactured in vivo.

**Table 1 jfb-13-00156-t001:** The clinical and surgical data of the patients from the tumour group.

Patients	1	2	3	4	5
Sex	Male	Male	Female	Female	Female
Other diseases	no	yes	yes	no	yes
Tumor location (and bone defect location)	parietal left	parietal right	frontal right	temp. right	parietal right
Macroscopic brain oedema	no	yes	no	yes	yes
WHO grade	1	2	1	2	1
Histology	P	A	F	A	CL
Recidive	no	no	no	no	no
Dura removal	yes	partial	yes	yes	yes
Bone invasion	yes	yes	yes	yes	yes
Subcutaneous tissue invasion	yes	no	yes	yes	no
Bone defect size [cm]	9 × 5	8 × 12	15 × 15	7 × 17	15 × 7
CSF leak	no	no	no	no	no
Postop. compl.	no	infection	no	no	no
Number of surgeries altogether	1	3	1	1	1
Number of reconstructions after first surgery	0	2	0	0	0
Approximate time for artificial flap manufacturing [minutes]	8	10	12	8	10
Recovery time/hospital stay after first operation [days]	6	7	7	5	7

Legend: M—meningothelial; Occipit.—occipital; A—atypical; F—fibroblastic; P—psammomatous; T—transitional; CL—clear cell; Postop. compl.—postoperative complication; CSF—cerebrospinal fluid.

**Table 2 jfb-13-00156-t002:** The clinical and surgical data of the trauma patients.

Patients	1	2	3	4
Sex	Male	Male	Male	Female
Other diseases	no	no	yes	no
Bone defect location	frontoparietal left	parietal right	frontoparietal right	temporoparietal right
Macroscopic brain oedema	severe	severe	severe	severe
Dural tear	yes	no	no	yes
Bone fragmentation	yes	no	no	yes
Bone defect size [cm]	15 × 20	15 × 25	18 × 25	20 × 25
Intraoperative CSF leak	yes	no	no	yes
Postop. compl.	no	no	no	no
Number of surgeries altogether	2	2	2	2
Number of reconstructions after first surgery	1	1	1	1
Approximate time for artificial flap manufacturing [minutes]	10	11	8	10
Recovery time/hospital stay after first operation [months]	4	3	6	4

**Table 3 jfb-13-00156-t003:** The clinical and surgical data of the vascular patients.

Patients	1	2	3
Sex	Male	Female	Female
Underlying pathology	ruptured arteriovenous malformation	ruptured internal carotid artery aneurysm	ruptured internal carotid artery aneurysm
Other diseases	yes	yes	no
Bone defect location	frontoparietal right	frontoparietal right	frontoparietal left
Macroscopic brain oedema	severe	severe	severe
Bone defect size [cm]	20 × 15	15 × 15	12 × 15
Postoperative CSF leak	no	no	no
Postop. compl.	no	no	no
Number of surgeries altogether	2	2	2
Number of reconstructions after first surgery	1	1	1
Approximate time for artificial flap manufacturing [minutes]	10	9	11
Recovery time/hospital stay after first operation [months]	2	3	3

## Data Availability

Data supporting reported results can be found at the archive of Department of neurosurgery, University Medical Centre Ljubljana, Slovenia.

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
