# Peer review of "Clinical Applications of Poly-Methyl-Methacrylate in Neurosurgery: The In Vivo Cranial Bone Reconstruction"

_jfb, 2022, doi:10.3390/jfb13030156_

Round 1
Reviewer 1 Report
The paper “Clinical applications of biopolymers in neurosurgery: the in vivo cranial bone reconstruction” has been carefully reviewed. In the paper, we share the clinical experience with poly-methyl-methacrylate for a novel in vivo bone defect closure in various neurosurgical operations.
There are several aspects that must be modified before publication:
Page 1, line 2, The first part of the title does not fit the content of the article. “Clinical applications of biopolymers in neurosurgery”. It's all about the PMMA.
Page 2, line 66-67. It's incorrect. Polylactic acid is PLA. PLLA is poly(L-lactide).
Page 2, line 86. The properties of poly-methyl-methacrylate, , section 2. It should be in the introduction since it is the state of the art
Page 3, line 139. Materials and methods. The section is incorrect. You must indicate, in addition to the method, the materials used and the supplier.
Page 5, figure 2. “The liquid and the dry components (powder). Indicate what is the liquid and the dry components.
Author Response
Dear Sir or Madam,
Thank you very much for your review and the comments, which were very valuable to us for the revision. We tried our best to revise the text according to the comments. We have indicated the issues we addressed with red font in the body of the article. Please find the detailed answers to your questions below. We hope that the revisions are satisfactory and you will find the article now suitable for publication in the Journal of functional biomaterials.
Reviewer 1
COMMENT 1. The paper “Clinical applications of biopolymers in neurosurgery: the in vivo cranial bone reconstruction” has been carefully reviewed. In the paper, we share the clinical experience with poly-methyl-methacrylate for a novel in vivo bone defect closure in various neurosurgical operations. There are several aspects that must be modified before publication: Page 1, line 2. The first part of the title does not fit the content of the article. “Clinical applications of biopolymers in neurosurgery”. It is all about the PMMA.
ANSWER 1. Thank you for this comment. The title has been modified in a more precise form: Clinical applications of poly-methyl-methacrylate in neurosurgery: the in vivo cranial bone reconstruction. We hope that the introduction is satisfactory now.
COMMENT 2. Page 2, lines 66-67. It is incorrect. Polylactic acid is PLA. PLLA is poly(L-lactide).
ANSWER 2. Thank you for this notice. This was corrected in the text. For such purposes, metal (titanium, tantalum, titanium and their alloys, magnesium and its alloys), ceramic, poly-methyl-methacrylate, poly-L-lactic acid (PLA), polylactide (PLLA) and polyetheretherketone (PEEK) are most often used [29, 32-34].
COMMENT 3. Page 2, line 86. The properties of poly-methyl-methacrylate, section 2. It should be in the introduction since it is the state of the art.
ANSWER 3. This part was translocated into the Introduction. Our aim in the article was also to discuss the material in question (the poly-methyl-methacrylate) and therefore we dedicated a special section to it. This part is now in the Introduction and is distinguished by a subtitle.
COMMENT 4. Page 3, line 139. Materials and methods. The section is incorrect. You must indicate, in addition to the method, the materials used and the supplier.
ANSWER 4. Thank you for this important reminder. We have added the material and its manufacturer to the article. The artificial bone was modelled from two-component poly-methyl-methacrylate material. We have utilised Palacos R+G, a high viscosity, radiopaque bone cement with gentamicin (Heraeus Medical, Germany). This part was corrected in the Materials and methods section.
COMMENT 5. Page 5, figure 2. “The liquid and the dry components (powder). Indicate what is the liquid and the dry components.
ANSWER 5. Thank you for this observation. In the figure, the dry component is located in the bowel on the right and the liquid component is stored in two glass vials on the right. To obtain a suitable consistency of the poly-methyl-methacrylate, both glass vials are normally needed. The liquid is added to the dry powder and then mixed and settled. We have indicated both of these components in Figure 2.
Figure 2. The poly-methyl-methacrylate components are sterile and individually packed. The liquid component is provided in a glass vial (arrow) and the dry component (powder) is stored in separate packaging. Both components are mixed to obtain a suitable consistency. On the left, the original bone flap is visible, which will be disposed of due to tumour invasion. A new, artificial bone flap will be formed from the poly-methyl-methacrylate.
Reviewer 2 Report
The title appears inadequate, it refers to biopolymers while, in reality, only PMMA is used
Line 19: "was not possible" are you sure it is the correct verb tense?
Line 20: “regeneration”. The phenomenon of regeneration in mammals is very rare due to the complexity of the tissue structure. The most common outcome of lesions with loss of substance is scarring with the phases described in the ordinary texts of surgical pathology. I would avoid using this term which, among other things, is in contradiction with the next sentence (line 22-22).
Line 40: insert references
Line 44 insert references
Line 46 it is a bit repetitive; you have already explained this point.
Line 76: again, the same concept is explained using different words.
Line 99, there is always an antibiotic inside.
Line 118 are suspensory dots a typo or a failure to revise the text?
Line 137 this sentence should be put in the introduction. Furthermore, I think that the explanation and history of PMMA should be summarised, it is a widely used and well-known material, so in my view, it is rather pointless to go on at such length.
Line 172-173 What the authors call sterile paper appears to be (figure 3A) the wrapping paper of PMMA. It is also unclear to me how the thickness of the flap obtained can be correctly assessed: in the case of neoplasia, there is very often hyperostosis, and therefore, the thickness of the bone flap removed may not be equal to that of physiological bone. Since PMMA is a malleable material without the use of suitable instruments (gauge, for example), it is difficult to be sure of obtaining a flap with an equal thickness at every point.
Line 372_374: “These requirement have also been met in our study” what other study are you referring to? A bibliographic citation is missing.
The introduction is quite repetitive in several parts. The same concept is repeated repeatedly and expressed in different words, but contentwise it does not vary. Therefore, I think it might be helpful to review the introduction by trying to express once and in a comprehensive manner what is to be expressed. Furthermore, it is not stated what the objective of this paper is.
The property of poly-methyl-methacylate: I believe that in a scientific case study this part is useless, it reports notes that can be found on many books and on the product leaflets
Results: except for the age range, all information on patients is lacking:
numbers, numbers per group
histological diagnosis of neoplasms
size of the bone defect
number of reconstructions performed immediately
number of reconstructions performed after time since the first surgery
summary table
statistics
CT examination only postoperative, there is no remote control to evaluate the bone-PMMA integration. The results section should be better structured: you should indicate specifically the patients enrolled, the type of neoplasm, whether or not it invaded the bone, the surface area of the bone defect, and the recovery time.
The discussion seems to be a small review of materials used for bone defects, thus lacking a critical analysis and a fundamental rationale as to why using PMMA is better than other materials.
Ultimately, the article needs careful revision by the authors. The simplicity and inexpensiveness of the procedure cannot be taken into account as the only factors that allow reconstructive surgery to be successful. The aesthetic aspect is closely related to the psychological sphere of the patients, which is why a study of the flap in advance and a correct flap preparation process are mandatory.
There is also a lack of important information, e.g. regarding the position of the bone defect, as I do not think it is always possible to model very complex structures' freehand'. This is why biomedical engineering and 3D printing are bringing considerable benefits to the medical field in recent years.
For all the reasons listed above the paper, in my opinion, cannot be accepted for publication and I strongly encourage the authors to a radical revision, with the repair of the deficiencies found and submission as a new paper.
Author Response
Dear Sir or Madam,
Thank you very much for your review and the comments, which were very valuable to us for the revision. We tried our best to revise the text according to the comments. We have indicated the issues we addressed with red font in the body of the article. Please find the detailed answers to your questions below. We hope that the revisions are satisfactory and you will find the article now suitable for publication in the Journal of functional biomaterials.
Reviewer 2
COMMENT 1. The title appears inadequate, it refers to biopolymers while, in reality, only PMMA is used.
ANSWER 1. Thank you for this comment. Other reviewers have also noted this inconsistency and we are very grateful. The title has been modified in a more precise form since the poly-methyl-methacrylate was used in our study: Clinical applications of poly-methyl-methacrylate in neurosurgery: the in vivo cranial bone reconstruction. We hope that the introduction is satisfactory now.
COMMENT 2. Line 19: "was not possible" are you sure it is the correct verb tense?
ANSWER 2. This part was corrected: This progress, however, has not been possible without the advances and improvements in basic science and in the new technology that had paved the path.
COMMENT 3. Line 20: “regeneration”. The phenomenon of regeneration in mammals is very rare due to the complexity of the tissue structure. The most common outcome of lesions with loss of substance is scarring with the phases described in the ordinary texts of surgical pathology. I would avoid using this term, which, among other things, is in contradiction with the next sentence (lines 22-22).
ANSWER 3. Thank you for this observation. In this part, we meant regeneration and not reparation, since the former process is the goal of the regenerative approaches in medicine. When we talk about reparation, the functional tissue is replaced with fibrous tissue. We agree with the respected reviewer that the process of regeneration is very complex and also not complete and achievable in many instances during healing. However, this is the goal of modern regenerative approaches in tissue engineering. We also agree that the most common outcome of lesions with loss of substance is scarring.
As advised, we have corrected this part in the text. We have omitted the term regeneration and used reparation (scarring) instead. It is more appropriate, indeed.
COMMENT 4. Line 40: insert references, Line 44 insert references, Line 46 is a bit repetitive; you have already explained this point.
ANSWER 4. Thank you. We have corrected these parts of the text as advised.
COMMENT 5. Line 76: again, the same concept is explained using different words.
ANSWER 5. This part has been addressed.
COMMENT 6. Line 99, there is always an antibiotic inside.
ANSWER 6. Sometimes, the PMMA comes without antibiotics, which may be added to the mixture separately. It is true, however, that the PMMA material for clinical use always incorporates antibiotics. We have corrected this part accordingly.
COMMENT 7. Line 118 are suspensory dots a typo or a failure to revise the text?
ANSWER 7. These are suspensory dots. We have omitted them.
COMMENT 8. Line 137: this sentence should be put in the introduction. Furthermore, I think that the explanation and history of PMMA should be summarised, it is a widely used and well-known material, so in my view, it is rather pointless to go on at such length.
ANSWER 8. Thank you for this observation. We have moved the last sentence in the Introduction. The summary of the PMMA was also translocated in the Introduction. Since other reviewers advised not to change it, we have shortened it a bit. In case the respected reviewer suggests omitting it completely, we are happy to do so.
COMMENT 9. Line 172-173. What the authors call sterile paper appears to be (figure 3A) the wrapping paper of PMMA. It is also unclear to me how the thickness of the flap obtained can be correctly assessed: in the case of neoplasia, there is very often hyperostosis, and therefore, the thickness of the bone flap removed may not be equal to that of physiological bone. Since PMMA is a malleable material without the use of suitable instruments (gauge, for example), it is difficult to be sure of obtaining a flap with an equal thickness at every point.
ANSWER 9. Thank you for this question. The sterile paper here was used to wrap the bone flap before the application of the PMMA to prevent the sticking of the material to the bone flap surface. Any material that can be wrapped on the bone and then separated easily from it and prevent sticking the PMMA can be used. In our instance, we have used the sterile paper that is prepared at the sterilisation unit of our medical centre. The PMMA does not contain sterile paper or any other material for separation.
The thickness of the original cranial bone flap was assessed before the modelling with the calliper. Then, the exact artificial PMMA flap in dimensions was manufactured. The PMMA is soft during the polymerisation phase and can be modelled as wished. It can be thinned, bent, and shaped according to the desired form; thicker parts of the product can be modelled as wished. Once hardened, the shaping is not possible anymore. During the resection phase of the tumour, for example of the meningioma infiltrated bone, the whole diseased bone flap is removed in one piece and with a safety margin. During every operation, the extent of the resection is labelled and the bone is cut away from the tumour or the thickened bone part so that the normal, physiological cranial bone is reached. The artificial bone flap is then modelled according to the excised cranial bone and its thickness is compared both to the original bone flap, which is removed, and to the healthy cranial bone. The PMMA flap can be thinned and bent as necessary, to fit the bone defect. The neuronavigation is of great benefit during the operation and is used for planning the resection and the safe margin location (1-3).
It is correct that the PMMA is a malleable material, which is its benefit. The PMMA mass can be thinned as necessary and the thickness of the emerging artificial flap is measured with a scale during the phases of its manufacturing. The template (the original cranial bone) is very welcome since it provides information about the thickness and curvature of the artificial flap. For that reason, we have used the original bone as a template and it was therefore not difficult to produce a correct shape.
These reviewer comments are important and we have included them in the manuscript text.
COMMENT 10. Line 372-374: “These requirements have also been met in our study” what other study are you referring to? A bibliographic citation is missing.
ANSWER 10. The citation has been added and the sentence rewritten (4-6). Here, we meant the PMMA properties.
COMMENT 11. The introduction is quite repetitive in several parts. The same concept is repeated repeatedly and expressed in different words, but contentwise it does not vary. Therefore, I think it might be helpful to review the introduction by trying to express once and in a comprehensive manner what is to be expressed. Furthermore, it is not stated what the objective of this paper is.
ANSWER 11. Thank you for this remark. The Introduction has been changed and rewritten. The objective of the study has been added: In the article, we share our clinical experience with poly-methyl-methacrylate for a novel in vivo bone defect closure in various neurosurgical operations.
COMMENT 12. The property of poly-methyl-methacrylate: I believe that in a scientific case study this part is useless, it reports notes that can be found in many books and the product leaflets.
ANSWER 12. We have shortened this part and since other reviewers agreed to include it in the text, we kindly ask the resected reviewer if it could stay.
COMMENT 13. Results: except for the age range, all information on patients is lacking:
numbers, numbers per group
histological diagnosis of neoplasms
size of the bone defect
number of reconstructions performed immediately
number of reconstructions performed after time since the first surgery
summary table
statistics
CT examination is only postoperative, there is no remote control to evaluate the bone-PMMA integration. The results section should be better structured: you should indicate specifically the patients enrolled, the type of neoplasm, whether or not it invaded the bone, the surface area of the bone defect, and the recovery time.
ANSWER 13. Thank you for this suggestion. We have added this information, as suggested, and tables summarising it.
COMMENT 14. The discussion seems to be a small review of materials used for bone defects, thus lacking a critical analysis and a fundamental rationale as to why using PMMA is better than other materials.
Ultimately, the article needs careful revision by the authors. The simplicity and inexpensiveness of the procedure cannot be taken into account as the only factors that allow reconstructive surgery to be successful. The aesthetic aspect is closely related to the psychological sphere of the patients, which is why a study of the flap in advance and a correct flap preparation process are mandatory.
There is also a lack of important information, e.g. regarding the position of the bone defect, as I do not think it is always possible to model very complex structures' freehand'. This is why biomedical engineering and 3D printing are bringing considerable benefits to the medical field in recent years.
ANSWER 14. Thank you for this suggestion. The Discussion has been supplemented as advised and other reasons for PMMA use in neurosurgery have been added.
The 3D-printing and computer-assisted manufacturing is indeed gaining importance. However, these new techniques also have some drawbacks, making the in vivo bone flap modelling a suitable alternative in comparison to more complicated techniques. Besides price, the 3D manufacturing requires time, first completing accurate imaging (usually a CT scan for a 3D reconstruction). Then, the manufacturing process itself is quite long. In some clinical settings and situations, it is not possible to wait for a manufactured 3D-bone flap and thus, in vivo modelling is preferable. Not all centres have access and the possibility to use such products. Additionally, the vital cranial bone near the bone defect (at the site of craniotomy) may change with time and since 3D manufacturing may be time-consuming, the artificial flap may not fit the bone defect optimally. During the in vivo modelling, on the other hand, the artificial PMMA flap can be adjusted and forms exactly according to the cranial bone defect. It is true, however, that all the defects of the skull bone cannot be covered. Especially problematic are those on the cranial base and the curvatures in the orbital area. These very complex structures cannot be modelled freehand and therefore, other reconstructive techniques must be used. The 3D printing represents considerable benefits for the reconstruction of such complex defects (7-9).
References
- Zamorano, L.; Kadi, A.; Dong, A. Computer-assisted neurosurgery: simulation and automation. Stereot Funct Neuros 1992, 59, 115-122.
- Fernández-de Thomas RJ, De Jesus O. Craniotomy. 2022. In: StatPearls [Internet]. Treasure Island (FL): StatPearls Publishing; 2022.
- Jean WC, Huang MC, Felbaum DR. Optimization of skull base exposure using navigation-integrated, virtual reality templates. J Clin Neurosci. 2020;80:125-130.
90 je 4. Wang Z, Li Z, Zhang X, Yu Y, Feng Q, Chen J, Xie W. A bone substitute composed of polymethyl-methacrylate bone cement and 571 Bio-Gene allogeneic bone promotes osteoblast viability, adhesion and differentiation. Biomed Mater Eng. 2021;32(1):29-37.
- Jäger M, Wilke A. Comprehensive biocompatibility testing of a new PMMA-hA bone cement versus conventional PMMA cement in vitro. J Biomater Sci Polym Ed. 2003;14(11):1283-98.
- Sa Y, Yang F, Wang Y, Wolke JGC, Jansen JA. Modifications of Poly(Methyl Methacrylate) Cement for Application in Orthopedic Surgery. Adv Exp Med Biol. 2018;1078:119-134.
- Scerrati A, Travaglini F, Gelmi CAE, Lombardo A, De Bonis P, Cavallo MA, Zamboni P. Patient-specific Polymethyl methacrylate customised cranioplasty using 3D printed silicone moulds: Technical note. Int J Med Robot. 2022;18(2):e2353.
- Đurić KS, Barić H, Domazet I, Barl P, Njirić N, Mrak G. Polymethylmethacrylate cranioplasty using low-cost customised 3D printed moulds for cranial defects - a single Centre experience: technical note. Br J Neurosurg. 2019;33(4):376-378.
- Lannon M, Algird A, Alsunbul W, Wang BH. Cost-Effective Cranioplasty Utilizing 3D Printed Molds: A Canadian Single-Center Experience. Can J Neurol Sci. 2022;49(2):196-202.
Reviewer 3 Report
The scientific paper "Clinical applications of biopolymers in neurosurgery: the in vivo cranial bone reconstruction” aimed to report clinical experience of a group of researchers with poly-methyl-methacrylate for a novel in vivo bone defect closure and artificial bone flap development in various neurosurgical operations. I can make the following considerations:
1) Authors should pay attention to the instructions to authors of the JFB journal, which are not adequate in the submitted manuscript. Since the title must contain capital letters, the affiliations of the authors is incomplete, the references outside the norms and more other errors. Please adjust.
2) The abstract is totally wrong. Too short without considering all the constituents of a scientific article: introduction, clear objective, methodology, results with discussion and conclusions.
3) In line 28, correct “O the other hand”
4) Separate the second paragraph that is too long (lines 24-45)
5) What does item 2: The properties of poly-methyl-methacrylate mean? Is it part of the Introduction? In fact, the introduction is very dispersed in relation to the main purpose of the manuscript. Authors should join item 2 to the introduction, reducing its content to be clearer, concise and objective. I suggest it be completely remodeled and rewritten.
6) The clinical study looks more like a case report, does not clearly demonstrate the criteria for inclusion of patients in the sample. How many patients were participants in the survey were made? How was your selection? Were comparisons made with control cases?
7) The images are poor, especially figure 2. Figures 4 and 5 could be together, in a single image.
8) The number of patients (20) is described in the results. It should be in the methodology with the selection criteria.
9) The discussion is more like an introduction because there are few relationships between the results obtained in the experiment and studies described in the literature.
Author Response
Dear Sir or Madam,
Thank you very much for your review and the comments, which were very valuable to us for the revision. We tried our best to revise the text according to the comments. We have indicated the issues we addressed with red font in the body of the article. Please find the detailed answers to your questions below. We hope that the revisions are satisfactory and you will find the article now suitable for publication in the Journal of functional biomaterials.
Reviewer 3
COMMENT 1. The scientific paper "Clinical applications of biopolymers in neurosurgery: the in vivo cranial bone reconstruction” aimed to report the clinical experience of a group of researchers with poly-methyl-methacrylate for a novel in vivo bone defect closure and artificial bone flap development in various neurosurgical operations. I can make the following considerations:
Authors should pay attention to the instructions to authors of the JFB journal, which are not adequate in the submitted manuscript. Since the title must contain capital letters, the affiliations of the authors are incomplete, the references are outside the norms and more other errors. Please adjust.
ANSWER 1. Thank you. This has been adjusted.
COMMENT 2. The abstract is totally wrong. Too short without considering all the constituents of a scientific article: introduction, a clear objective, methodology, results with discussion and conclusions.
ANSWER 2. The abstract has been rewritten.
COMMENT 3. In line 28, correct “O the other hand.”
ANSWER 3. Thank you. This was corrected.
COMMENT 4. Separate the second paragraph that is too long (lines 24-45).
ANSWER 4. The paragraph was separated and shortened.
COMMENT 5. What does item 2: The properties of poly-methyl-methacrylate mean? Is it part of the Introduction? The introduction is very dispersed in relation to the main purpose of the manuscript. Authors should join item 2 to the introduction, reducing its content to be clearer, concise and objective. I suggest it be completely remodelled and rewritten.
ANSWER 5. Thank you for this remark. The paragraph under his title is a part of the Introduction and we have rewritten and changed the whole part. We hope that is satisfactory now.
COMMENT 6. The clinical study looks more like a case report, does not clearly demonstrate the criteria for inclusion of patients in the sample. How many patients were participants in the survey were made? How was your selection? Were comparisons made with control cases?
ANSWER 6. Thank you for this observation. In this study, we wanted to expose the technical part of the reconstruction procedure, with the emphasis on the material, its properties and the presentation of the in vivo reconstructive technique. We have added the inclusion criteria and the number of patients. We did not have control cases, as it was not possible to have any in a such retrospective review. We reported our experience with the technique and the material.
COMMENT 7. The images are poor, especially in figure 2. Figures 4 and 5 could be together, in a single image.
ANSWER 7. We have joined Figures 4 and 5. Figure 2 is meant to demonstrate the liquid and powder components of the PMMA. We ask the resected reviewer if it can remain.
COMMENT 8. The number of patients (20) is described in the results. It should be in the methodology with the selection criteria.
ANSWER 8. Thank you for this observation. This was corrected as advised.
COMMENT 9. The discussion is more like an introduction because there are few relationships between the results obtained in the experiment and studies described in the literature.
ANSWER 8. Thank you for this suggestion. The Discussion has been rewritten as advised.
Round 2
Reviewer 2 Report
After the modifications made by the authors, the article appears much more complete and interesting.
There is still a small observation:
187: ... Hunt Hess ...: requires a bibliographic reference
Author Response
Dear Sir or Madam,
Thank you very much for your second review and the comments. We have corrected and added the missing parts, the reference especially. We hope that the revisions are satisfactory and you will find the article now suitable for publication in the Journal of functional biomaterials.
Reviewer 1
COMMENT 1. After the modifications made by the authors, the article appears much more complete and interesting. There is still a small observation: 187: ... Hunt Hess ...: requires a bibliographic reference.
ANSWER 1. Thank you for this kind comment. We have added the missing references in the text. They are labelled in red.
Reviewer 2
COMMENT 1. No comments.
ANSWER 1. Thank you for his kind comment.
Reviewer 3 Report
No comments
Author Response

(The authors gave the same response as above.)
